# Effects of Wetting and Drying Cycles on Microstructure Change and Mechanical Properties of Coconut Fibre-Reinforced Mortar

**Huyen Bui [1], Daniel Levacher [2,\*], Mohamed Boutouil [3] and Nassim Sebaibi [3]**

[1]  Unité de Chimie Environnementale et Interactions sur le Vivant, Littoral Côte d'Opale Université, UR 4492, UCEIV, SFR Condorcet FR CNRS 3417, 145 Avenue Maurice Schumann, 59140 Dunkerque, France; thi-thu-huyen.bui@univ-littoral.fr

[2]  Faculty of Sciences, ComUE Normandie Université, M2C UMR 6143 CNRS, Unicaen, 24 Rue des Tilleuls, 14000 Caen, France

[3]  Laboratoire ESITC-ESITC Caen, ComUE Normandie Université, 14160 Epron, France; mohamed.boutouil@esitc-caen.fr (M.B.); nassim.sebaibi@esitc-caen.fr (N.S.)

\*  Correspondence: daniel.levacher@unicaen.fr

**Abstract:** Natural fibre-reinforced cementitious composites are commonly used as outer construction materials. They usually suffer weather as a result of being expose to various types of climates. In this study, a series of experimental tests were carried out to investigate the deterioration mechanism and mechanical properties of mortars incorporating coconut fibres due to repeated wetting and drying. The results indicated that although the compressive strength was found to increase after the first cycle, both compressive and flexural strengths underwent a significant decrease in the fifth cycle. In addition, at high temperatures, mortar matrixes retain their stable structure, according to the results of TGA analysis. When wetting and drying curing was applied, there was a significant degradation of fibres in the mortar.

**Keywords:** coconut fibre; wetting and drying cycle; moisture absorption; mechanical properties; thermogravimetric analysis

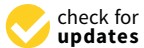

## 1. Introduction

Weather changes are one of the most detrimental external factors in the long-term survival of reinforced-composite materials. Many attempts [1–5] have been made to investigate weathering effects on reinforced-composites for long-term structural applications, because most of these materials are applied for outer construction. Several studies have shown that when natural fibres were incorporated, composites were sensitive to high humidity and temperature compared to man-made fibres because of their hydrophilic properties and thermal sensitivity [6–8]. Natural fibre-reinforced cementitious composites are mainly used in outer applications, where the environment is considered as an aggressive factor. Therefore, these composite materials suffer losses in strength and decrease in durability. It is necessary to evaluate the durability in the face of wetting and drying exposure of cementitious composites in order to prevent and mitigate composite degradation. Parameters considered in previous studies on the effects of cyclic wetting and drying are listed in Table 1, including experimental conditions and objectives.

The mechanical properties of composite materials need to sound, considering the environmental vulnerability. The mechanical properties of fibre-reinforced composite resulting from wetting and drying cycles have been investigated in several previous studies [9,10]. Generally, exposure to wetting and drying cycles has a strong effect on mechanical properties of samples due to the repetition of negative environmental factors on the interfacial bonding between fibres and the cementitious matrix [11,12]. After the sample is exposed to wetting and drying cycles, compressive strength is the most

critical factor in assessing the performance of composite materials [13]. For instance, the decrease in strength of kraft pulp fibre-reinforced cement paste was observed in the study of Mohr et al. [14]. After 25 cycles of fresh-water exposure at ambient temperature, the material retained only 1–2% of post-cracking toughness. The authors also explained three development steps of composite degradation under repeated wetting and drying. First, debonding between fibres and the cementitious matrix occurs within the first two cycles because of drying shrinkage of fibres, followed by a remarkable loss in mechanical properties of composite and dimensional change of fibres due to the production of hydration products in the pores at the fibre–cement interface. This stage takes place prior to the 10th cycle, when fibres are mineralized. In this last stage, although a negligible increase in strength is obtained, toughness cannot be not regained.

Similarly, the compressive strength of alkali-activated composite materials showed a significant downward trend when these specimens were placed in a wetting and drying environment [15]. This strength reduction could be explained by several reasons. Firstly, the shrinkage of the specimen in the drying stage induces the formation of cracks and subsequently decreases the strength. In addition, the degradation of the microstructure due to alkali attack also leads to seriously reduced mechanical performance. Sodium solution crystallizes in the porosity space and creates inner pressure to contribute to strength decline. The final reason is due to alkali leaching, which restrains alkali activation and likely diminishes strength. By contrast, the mechanical properties seem to be not affected significantly by several wetting and drying cycles, according to Sodoke et al. [6]. Observation of the relationship between stress and strain of cycle flax/epoxy composite indicated that an insignificant decrease in the initial slopes, i.e., deformation modulus, was obtained after a 104-day period of exposure, due to the crystallization effect of fibre treatment. Meanwhile, the deterioration process of mortar cement kept in a saltwater environment was diminished due to the delay of chloride diffusion in the porosity of the matrix [16]. Before decreasing slightly in the 120th cycle, the relative dynamic modulus of elasticity (deduced from an ultrasonic pulse velocity (UPV) test) of the mortar had an upward trend during the first 90 cycles. In addition, the mass of the mortar increased to 2.74% after the exposure period compared to the control mortar (unaged cycling). Consumption water during the hydration process causes the formation of unsaturated pore space, and thus the mortar specimen continues to absorb water and increase its mass until it reaches maximum water saturation [17,18].

Many investigations on the effects of weathering on the long-term properties of composite materials have been performed concerning durability [19], degradation mechanisms [13,16,20], microstructure and composition change [18,21], and mechanical performance [22–26]. However, most of the investigations focus on conventional concrete, while knowledge of natural fibre-reinforced composites is still limited. Thus, understanding of the effects of wetting and drying exposure on the microstructure change and mechanical properties of natural fibre-reinforced mortar is essential and contributes to (i) the control of the change in the microstructure and (ii) the improvement of the mechanical properties of mortars.

**Table 1.** Literature review addressing the influences of wetting and drying cycles on the multi-physical and mechanical properties of different composite materials.

| Ref. | Type of Composite Materials | Treatment Method | | Properties Investigated | The Number of Cycles, n (Maximum) | Parameters Tested | Variation Trend with n |
|---|---|---|---|---|---|---|---|
| | | Wetting | Drying | | | | |
| Sodoke et al. [6] | Flax/epoxy composites | IWW until saturation | OD at 60 °C for 48 h | Mechanical and physicochemical properties | 7 | Water absorption $W_{ab}$ | -$W_{ab}$ decreases slightly -WD cycles have no severe effects in reducing mechanical properties Good retention of mechanical properties -Thermal degradation is more resilient after WD |
| Yin et al. [11] | Textile-reinforced concrete column | | | | 90 | | -Bearing capacity and ductility decrease |
| Wei et al. [12] | Basalt fibre-reinforced polymer concrete | Immersed in 3.5% NaCl solution for 8 h at 40 °C | AD at 25 °C for 16 h | Bond-slip behaviour | 360 | | -Weaker and more brittle |
| Mohr et al. [19] | Pulp fibre–cement composite | IWW at 65 °C during 23.5 h, AD at 22 °C for 30 min | OD at 65 °C for 23.5 h, AD at 22 °C for 30 min | Flexural properties | 25 | First crack strength Peak strength Post-cracking toughness | -Decrease significantly |
| Yin et al. [26] | Textile-reinforced concrete thin plate | Immersed in 5% NaCl solution for 12 h | AD for 12 h | Mechanical properties | 150 | Interfacial bonding strength between yarn fibres and fine-grained concrete | -Decrease -Mechanical properties have not been significantly improved -Deterioration increases |
| Mohr et al. [27] | | | | Microstructural and chemical properties | | | |
| ASTM 4843 | Solid wastes | IWW for 23 h | OD at 60 °C for 24 h | | 12 | | |

Note: IWW: immersed in water for wetting; OD: oven drying; AD: air drying; WD: wetting-drying.

## 2. Materials and Methods

### 2.1. Materials

In this study, coconut fibres originating from Vietnam were used in the form of natural fibres. The average density and diameter of fibre were 1.14 g/cm$^3$ and 250 μm, respectively, while the maximal water absorption after a two-day immersion period was 133% and the length was 1–2 cm. Concerning mechanical properties, several experiments were carried out, determining that the coconut fibre had low tensile strength (only 123 MPa in comparison with the value of 400–800 MPa of others) and high strain (up to 27% compared to the value of 3–5% of other natural fibres). Based on a survey of three hundred fibres using microscope measurements, the length and diameter of fibres are shown in Figures 1 and 2, respectively.

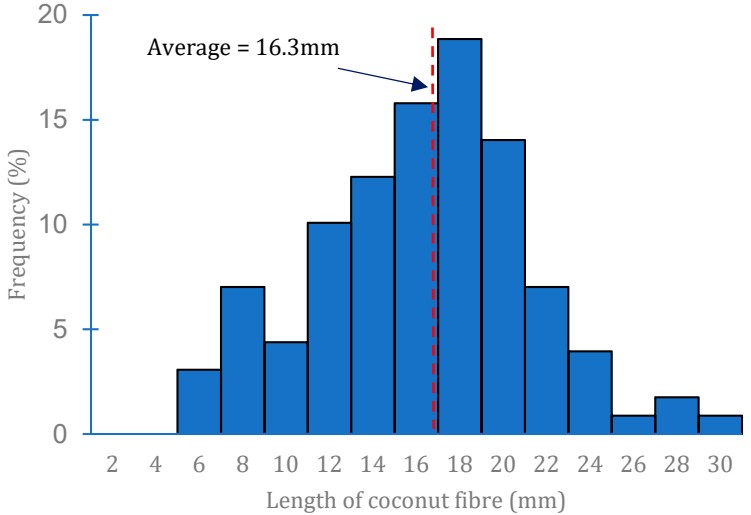

**Figure 1.** Length distribution of coconut fibres.

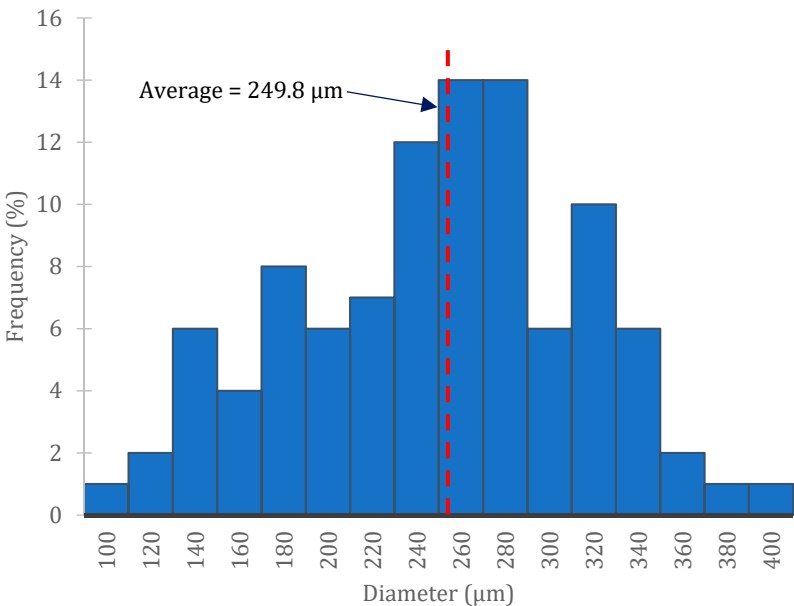

**Figure 2.** Diameter distribution of fibres.

Two types of cement, including Portland cement CEM I 52.5 N (PC) and Calcium Sulfoaluminate cement (CSA cement), were used to manufacture the mortar samples. The mortar specimens were produced in accordance with the EN 196-1 standard. Fibres were incorporated into the mortar at levels of 0% (without fibres), 1%, 2% and 3%, with the

percent expressed in volume of mortar, corresponding the sample symbols of PC, PC1, PC2 and PC3 for Portland cement and CSA, CSA1, CSA2 and CSA3 for CSA cement, respectively. The addition of fibres consisted of replacing a volume of sand with a corresponding volume of fibre, with a constant total mass of fibres and sand of 1350 g. The samples were cast and cured in the laboratory, as shown in Figure 3. The selection and specifications of raw materials and mortar manufacturing are presented in detail in previous studies [27–29].

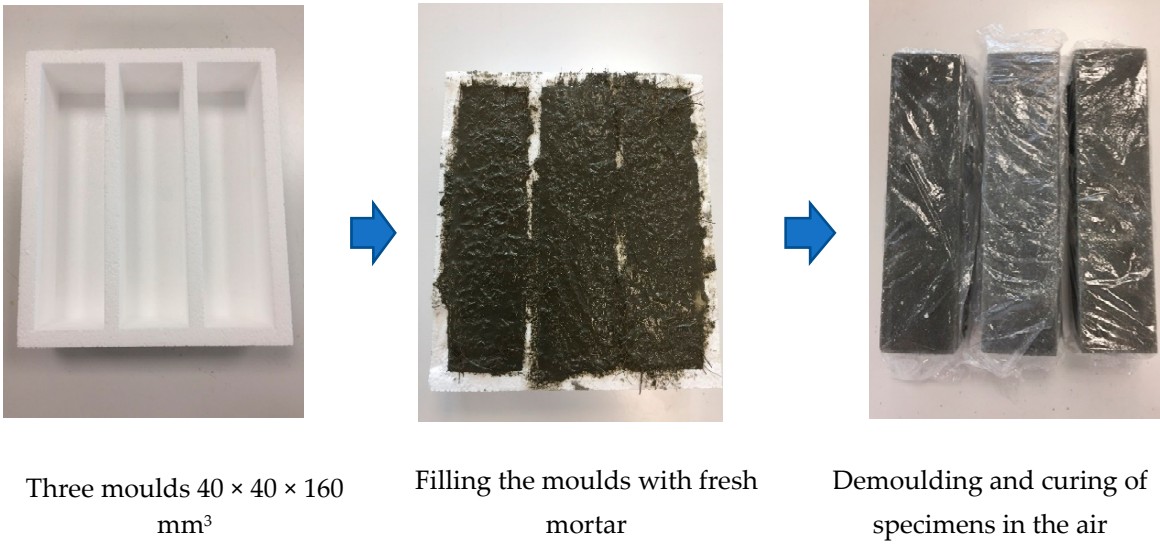

| Three moulds 40 × 40 × 160 mm³ | Filling the moulds with fresh mortar | Demoulding and curing of specimens in the air |

**Figure 3.** The casting and curing of samples.

*2.2. Wetting and Drying Exposure Conditions*

A wetting and drying cycle is defined in Figure 4 as the following: 24 h drying in a ventilated oven at 55 °C and 24 h immersion in water in ambient conditions. Before each exposure (immersion or drying), the sample was stored in air for at least 1 h to dry or cool down at the ambient temperature to avoid undesirable micro-cracks due to thermal shock. Simultaneously, the mass of the sample was measured to allow measurement of change (loss/gain) during wetting and drying. This ensured stability, which means that the change in the mass of sample was less than 1% within 2 h before conducting the next cycles. Samples were exposed during 1 cycle, i.e., 2 days, and 5 cycles, i.e., 10 days, before testing. The properties of exposed and unexposed samples were evaluated, and the results were compared.

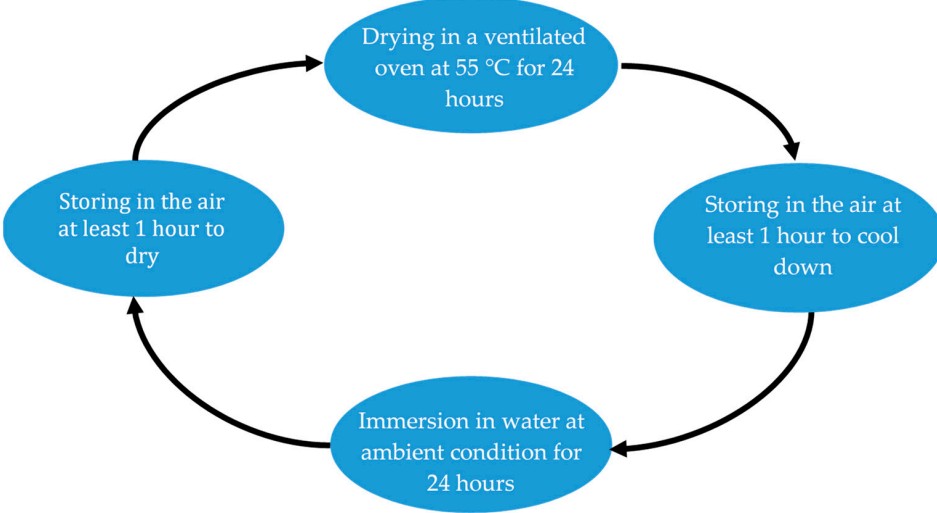

**Figure 4.** Wetting and drying cycle.

After different cycles of wetting and drying, the water absorption properties of mortars, evaluated by weight measurements and moisture absorption, were defined using $Ab_t$, according to the following equation:

$$Ab_t = \frac{W_w - W_d}{W_d} \times 100\% \tag{1}$$

where $Ab_t$ is the moisture absorption of mortar after exposure to $t$ cycles, in %; $W_w$, the mass of mortar after wetting, in g; and $W_d$, the mass of mortar after drying, in g. The value of moisture absorption was calculated based on an average of at least three specimens.

### 2.3. Mechanical Strength Testing

The mechanical strength of the mortars was determined in accordance with the NF EN 196-1 standard. In bending, tests were performed on hardened $40 \times 40 \times 160$ mm$^3$ prisms by means of an electromechanical press. A linear variable differential transformer (LVDT) was attached to the machine to record deformation for each sample tested. Three specimens were tested after curing time at a constant rate of deformation of 0.3 mm/min with a capacity of 50 kN load cell. The displacement at the mid-length of the prism and the flexural load were simultaneously recorded during the test to obtain the stress–strain relationship.

A compression test was carried out on two halves of the prism, broken after the bending test, i.e., six specimens. The same press machine was used. However, a load cell of 250 kN was used. As recommended by the NF EN 196-1 standard, the loading rate applied was 2.5 kN/s. The half-prism tested was placed laterally in the centre of the bearing plate of the machine. The mechanical test is shown in Figure 5.

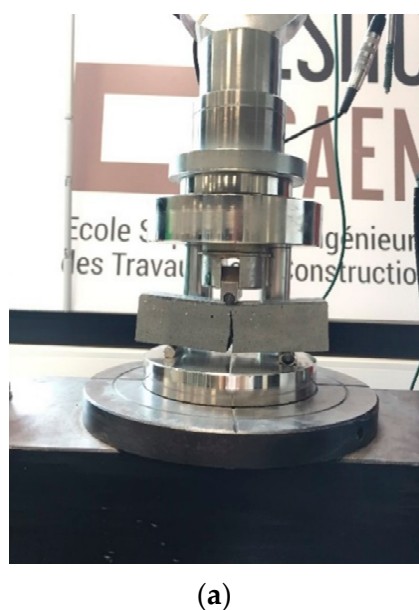
(**a**)

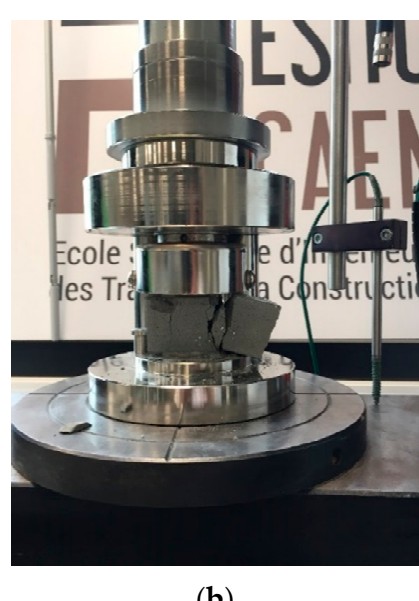
(**b**)

**Figure 5.** Mechanical test equipment. (**a**) Flexural three-point test. (**b**) Compression test.

### 2.4. TG-DTG Measurement

During the sample preparation process for TG-DTG analysis, samples were ground into powder. Thermal gravimetric analyses of mortar powders with and without exposure were conducted on a thermal analyser (STA 449 F5, NETZSCH-Gerätebau GmbH-Germany), as shown in Figure 6, in the temperature range of 20 °C–900 °C in a nitrogen atmosphere with a speed of 50 mL/min. The rate of heating was kept constant at 20 °C/min. Sample mass between 50 mg and 60 mg was put in a platinum crucible for these measurements.

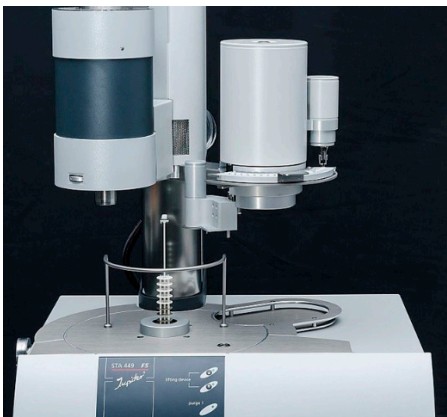

**Figure 6.** STA 449 F5 Jupiter used for thermal analysis.

## 3. Results and Discussion

### 3.1. Water Absorption

The effects of the addition of natural fibres and cyclic wetting and drying on the water absorption of mortars are shown in Figure 7. According to the results, there was a significant increase in water absorption with cycle number. Generally, the water absorption rate of PC-based mortars was higher than those of CSA cement-based mortars, regardless of fibre content and cyclic wetting and drying number. For instance, in the case of applying the wetting and drying environmental conditions, after the first cycle, the average values were around 3.4% and 2.6% for PC-based and CSA-base mortars, respectively. During this cycle, when the cement hydration process continued, the porosities and macro pores were filled by the hydration products formed due to the ye'elimite reactions in the CSA cement [30], leading the mortar to become denser. This phenomenon was responsible for the lower absorption of CSA cement-based mortar compared to PC-based mortar. After the hydration process was completed totally, the number of macro pores in the CSA cement-based mortar increased. One possible explanation for this may be that the voids were divided by numerous hydration products, creating larger pores during the hydration process. During this process, these voids could not be filled totally by the later hydration production, i.e., ettringite (AFt) and aluminium hydroxide (AH3), leading to a higher aggregation of pores [31]. Therefore, after five cycles, the moisture absorption of mortars based on CSA cement increased significantly. The dash lines in Figure 4 indicate that PC and CSA cement-based mortars under five exposure cycles exhibited moisture absorption nearly twice and three times as high as those at the first cycle, respectively. In addition, incorporating natural fibres into mortars also led to an increase in the voids surrounding fibres, resulting in higher absorption ability compared to unreinforced mortars.

### 3.2. Mechanical Properties

Samples without fibres, i.e., PC and CSA, and with 2% fibres, i.e., PC2 and CSA2, were chosen for mechanical investigation. This was because PC2 and CSA2 samples contain optimal fibre content for improvement of mechanical properties, according to a previous study [29]. For the sake of comparison, PC and CSA samples were also considered. In order to observe the damage progression and explain the influence of wetting–drying action, the mechanical tests were performed after 0, 1 and 5 cycles. Control samples, which were not exposed, were considered as reference samples. Data results of unconfined compressive and flexural strengths after cyclic wetting and drying are presented in Figure 8. The wetting and drying repeating has adverse effects on the mechanical performance of mortar, regardless of the number of fibres. Reducing of both strengths was observed. Generally, losses in mechanical properties of the CSA-based mortars were greater than those of PC-based mortars. However, the results highlighted that the maximum compressive strength was observed after one cycle, since complete hydration of the cement was reached

due to the supplementation of water during the wetting process. In next cycles, because of the formation of crystallised hydrate products [32], micro-cracks appeared gradually inside the mortar structure and induced a decrease in compressive strength. Both strength and deformation of the mortar samples decreased at the higher level of porosity and the higher number of cycles. The loss of strength was observed when fibres were incorporated into mortar. More pores in fine aggregate mortar appeared due to the adding of coconut fibres, causing a convenient environment for the deep penetration of ambient air and water. The change in mechanical strength with the predicted tendency was governed by the porosity, the number of cycles and the fibre content, i.e., the higher the fibre content, the higher the porosity and the higher the number of wet and dry cycles, the lower the mechanical strength.

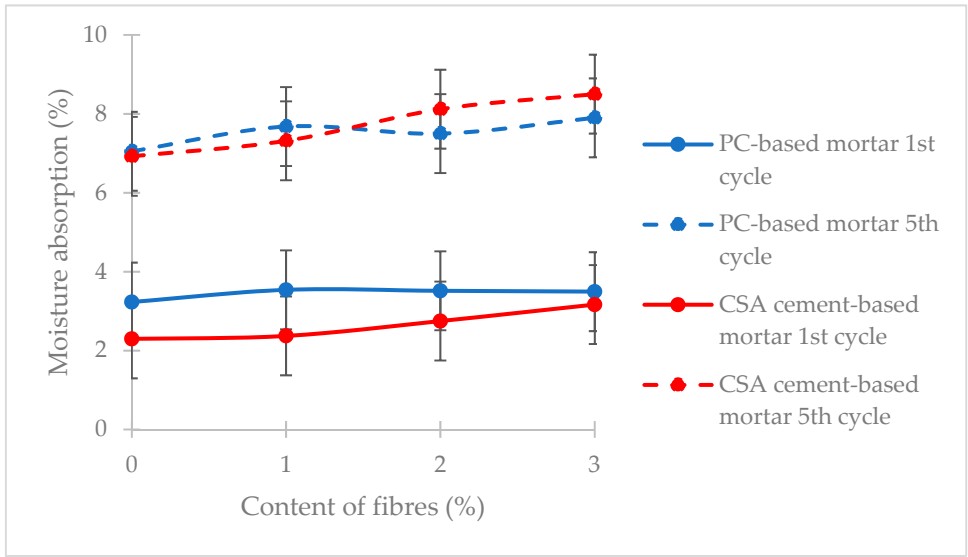

**Figure 7.** Moisture absorption versus fibre content for mixes.

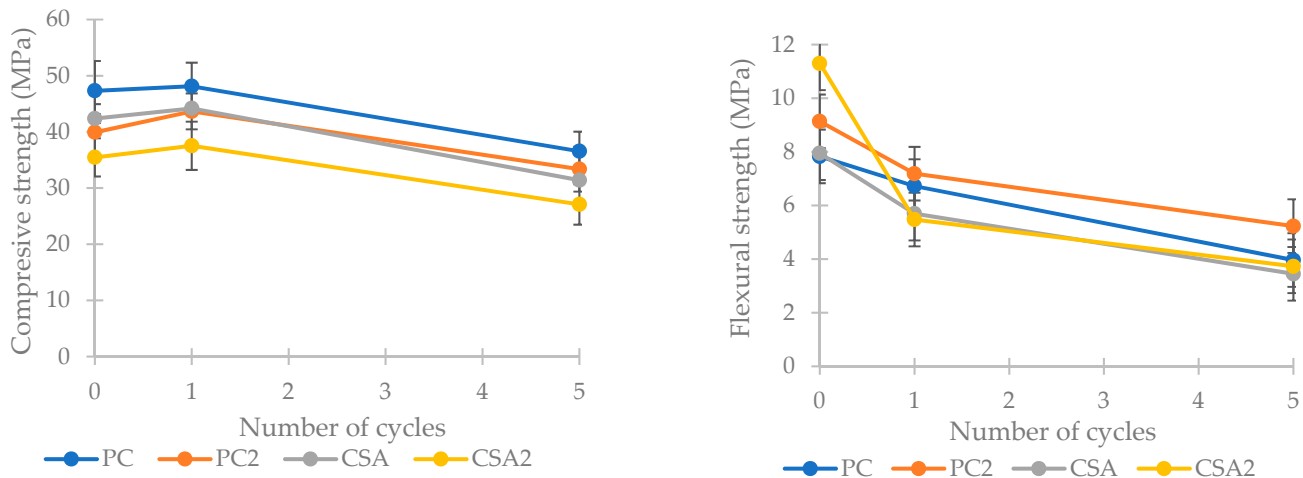

**Figure 8.** Compressive (**left**) and flexural (**right**) strengths of mortars after wetting and drying cycles.

The same trend was observed in bending: the strength of mortars decreased continuously from the first cycle. In comparison with the compressive strength reduction, the decrease of flexural strength occurred more quickly during the first cycle, with a dangerously high loss rate, shown by the steeper slope of strength reduction. Tang et al. [33] also confirmed that flexural strength loss was more severe than that of compressive strength under wetting and drying exposure conditions. The main reason is that the natural degradation

of fibres after wetting and drying exposure causes breaking of the bonding between fibres and the matrix, contributing to the reduction of the flexural strength of fibre reinforced-cementitious materials. Consequently, the bonding properties deteriorated, which led to the rapid decrease in the flexural performance of the specimens. When the fibres were added, the effects of fibre degradation on mortar damage were stronger than that of the bridging effect of the fibre distribution.

During the bending test, the deformation of samples was continuously observed, and the displacement was measured. The typical load–displacement relationships of mortars after five cycles compared to the reference sample are illustrated in Figure 9. As can be observed, the control fibre-reinforced mortar showed well-known behaviour and a good performance in preventing the sudden brittle fracture due to fibre pull-out after the first appearance of cracking. Meanwhile, a significant decrease in both maximum flexural strength and displacement was clearly observed in the wake of repeated wetting and drying. The mortar sample exposed to five cycles failed rapidly, while a significantly higher displacement value was observed for the reference sample. As shown in the force–displacement curve, after five cycles of wetting and drying, the samples presented a remarkable reduction in peak strength and post-cracking. The same observation was reported in previous studies [24,25]. After several cycles, the degradation of fibres made their ductility inadequate for crack-bridging capacity, and sequences of this fibre deterioration caused displacement reduction. This could also be related to the crack formation due to shrinkage in the drying process and formation of hydration products, i.e., ettringite, in the wetting process [25]. The formation of shrinkage-induced cracking usually occurred due to the removal of adsorbed water from the cementitious matrix during the hydration process [34].

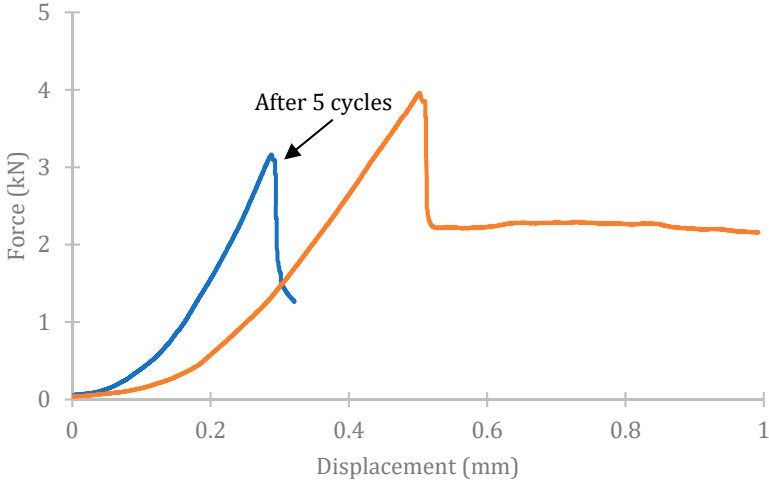

**Figure 9.** Typical force–displacement relationship of mortar samples incorporating fibres.

### 3.3. Thermogravimetric Analysis

The results of TG and DTG analysis of fibre-reinforced mortar prepared with PC and CSA cement, including 2% fibres and without fibres, after five wetting and drying cycles are plotted in Figure 10. The weight loss along different temperature ranges is identified in Figure 11. Concerning the thermal performance, the main three decomposition peaks, indicated in three dash lines, were clearly observed. They corresponded to evaporation and C-S-H, $Ca(OH)_2$ and $CaCO_3$ decomposition, according to three temperature ranges of 80–110 °C, 450–480 °C and 750–800 °C, respectively. It should be noted that for reference samples, i.e., without any cyclic wetting and drying, the fourth peak was obtained at a temperature of 270 to 300 °C. This decomposition peak concerned the cellulose of the fibre. It seems to have disappeared for the samples that underwent several cycles of wetting and drying due to the drawback of the degradation of fibres, i.e., the PC2–5 cycle and CSA2–5 cycle.

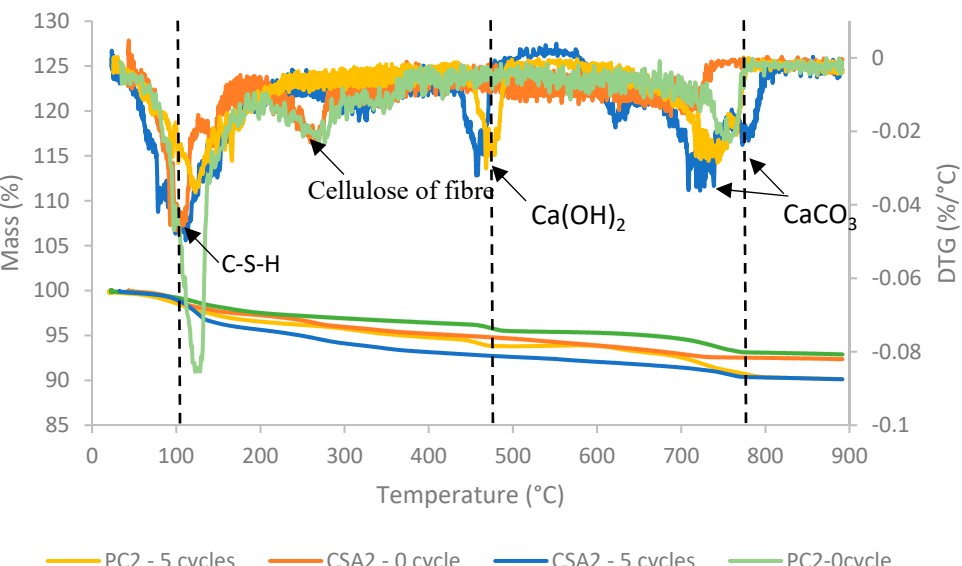

**Figure 10.** TG and DTG graphs for mortar incorporating fibres after 0 and 5 wetting and drying cycles.

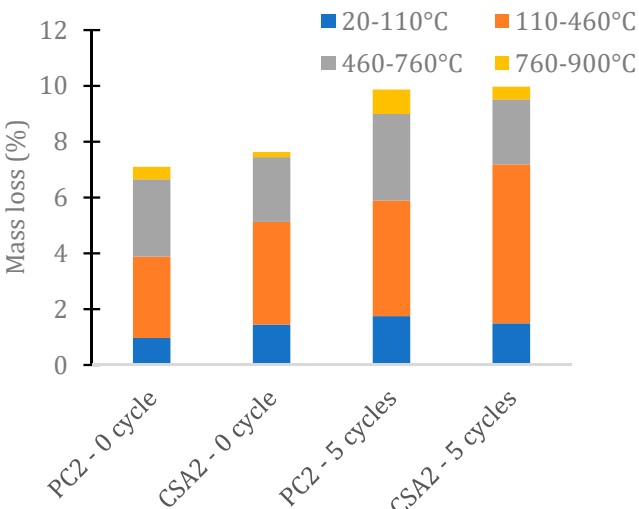

**Figure 11.** Mass loss for mortar samples incorporating fibres after 0 and 5 wetting and drying cycles in transition temperature ranges.

In terms of mass loss, the pyrolysis behaviours of the four samples analyzed had the same trends once mass loss fractions occurred continuously in accordance with the increasing temperature. Four main steps were identified in the degradation process due to the endothermal effects, including (I) 20–110 °C (evaporation and C-S-H), (II) 110–460 °C (C-S-H and cellulose), (III) 460–760 °C (portlandite and poor-crystallized carbonates) and (IV) 760–900 °C (well-crystallized carbonates). The mass loss level increased gradually and then rapidly moved to a stable trend at approximately 800 °C. For instance, during the first of the decomposition periods, i.e., from room temperature to 110 °C, the reference and wetting and drying samples thermally behaved very similarly; no difference in the resistance to temperature changes of the four samples was observed. During this temperature range, the mass loss fraction was less than 2% for the four samples tested. From 110 °C, the slope of the mass fractions of the samples under wetting and drying was steeper compared to that of reference samples. Among the steps of degradation, the strongest deterioration took place in the two intermediate steps, in which 6–8% of the mass loss was reported. At the end of the decomposition process, while the PC2–0 cycle and CSA2–0 cycle samples still retained approximately 93% of their mass, the two other ones did not weigh more than 90%

of their initial weight. As a result of thermal analysis, the reference samples had a better ability to resist temperature than the samples after wetting and drying cycles.

## 4. Conclusions

Natural fibre-reinforced composite building materials are usually subjected to environmental vulnerability due to weathering changes. In this study, the mechanical properties and microstructure changes of a cementitious matrix prepared with two types of cement, i.e., PC and CSA cement, and reinforced with coconut fibres at different content levels was investigated experimentally. The results were compared to those of mortars without fibres and those not subjected to wetting and drying exposure. According to the measurement results, the following conclusions can be drawn:

-   The effects of cyclic wetting and drying on the absorption capacity of mortars were clearly observed, since the moisture absorption of the PC-based and CSA cement-based mortars at the fifth cycle were twice and triple those of the first cycle, respectively. In addition, when fibres were incorporated, a high absorption ability was found due to the increase in the voids surrounding the fibres.
-   The mechanical strength results presented a significant decrease in the fifth cycle in all mixtures in comparison with the reference sample. However, due to the total hydration of cement, a slight increase in compressive strength was observed after the first cycle. The loss in flexural strength is more remarkable than that in compressive strength. The influence of fibre degradation on sample damage also dominates the bridging effect of fibre distribution resulting from wetting and drying exposure. The mechanical strength loss of CSA cement-based mortars was considered to be greater than that of PC-based mortars.
-   The natural degradation of fibres in the cementitious matrix after five cycles occurred, shown by observation of no decomposition peaks at a temperature range of 270 to 300 °C in PC2–5 and CSA2–5 samples. The mass loss with the temperature of sample under natural curing was found to be slightly lower than that of the sample applied to wetting and drying curing, regardless of fibre addition.

In general, plant fibre-reinforced cementitious composites should be utilized in interior parts of building construction, such as panels, walls, ceiling, dashboard parts, etc. The long-term durability of composites containing agricultural by-products should be considered for future studies to assess the potential application for the exterior parts of construction engineering.

**Author Contributions:** Conceptualization, H.B.; methodology and experimental work H.B.; formal analysis, H.B.; writing—original draft preparation, H.B.; writing—review and editing, D.L. and H.B.; supervision, M.B., N.S. and D.L.; project administration, M.B. and N.S. All authors have read and agreed to the published version of the manuscript.

**Funding:** This research received no external funding.

**Institutional Review Board Statement:** Not applicable.

**Informed Consent Statement:** Not applicable.

**Acknowledgments:** The first author deeply appreciates the Vietnamese government-911 scholarship, offering years of financial support for scientific learning and research study in ESITC Caen, France.

**Conflicts of Interest:** The authors declare that they have no known competing financial interest or personal relationships that could have appeared to influence the work reported in this paper.

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
