# Peer review of "Effects of Wetting and Drying Cycles on Microstructure Change and Mechanical Properties of Coconut Fibre-Reinforced Mortar"

_jcs, doi:10.3390/jcs6040102_

Round 1
Reviewer 1 Report
More detail information about physical and mechanical properties of fibers, that surely affect the resulting behavior of the composites, are missing.
Chapters 2.3. and 2.4. should contain more detailed description, preferably with graphical representation of the method used during the research.
The authors should clearly describe the advantages and disadvantages of these fibers, in regard to the other natural and artificial fibers.
The information about the behavior of developed samples in practical conditions are missing, especially information about the temperature.
Figure 7. needs modification to make the curves thinner.
The results should be more transparent. I suggest presenting advantages and disadvantages per sample, in a table form in Results and discussion.
The Conclusion should contain information on when to use these components and why, as well as information when the usage of these components is not recommended.
Reviewer 2 Report
The fiber micrograph must be presented.
The details of sample preparation and pictures of concrete/mortar samples should be provided.
What are the standards used for measurement of compression and flexural properties?
The presented results need to be explained in terms of statistical analysis to show the significance of the results. The error bars/SD should be mentioned.
Recent literature in the field should be cited. The novelty of the work should be enhanced by using SEM/ microscopy and analysis of weatharability of the samples.
The title says microstructure; therefore, microstructure of fiber and its interface must be included.
Round 2
Reviewer 2 Report
Can be accepted